# Brain representations of motion and position in the double-drift illusion

**Noah J Steinberg[1]\*[†], Zvi N Roth[1,2]\*[†], J Anthony Movshon[3], Elisha Merriam[1]**

[1]Laboratory of Brain and Cognition, National Institute of Mental Health, Bethesda, United States; [2]School of Psychological Sciences, Faculty of Social Sciences, Tel Aviv University, Tel Aviv, Israel; [3]Center for Neural Science, New York University, New York, United States

**Abstract** In the 'double-drift' illusion, local motion within a window moving in the periphery of the visual field alters the window's perceived path. The illusion is strong even when the eyes track a target whose motion matches the window so that the stimulus remains stable on the retina. This implies that the illusion involves the integration of retinal signals with non-retinal eye-movement signals. To identify where in the brain this integration occurs, we measured BOLD fMRI responses in visual cortex while subjects experienced the double-drift illusion. We then used a combination of univariate and multivariate decoding analyses to identify (1) which brain areas were sensitive to the illusion and (2) whether these brain areas contained information about the illusory stimulus trajectory. We identified a number of cortical areas that responded more strongly during the illusion than a control condition that was matched for low-level stimulus properties. Only in area hMT+ was it possible to decode the illusory trajectory. We additionally performed a number of important controls that rule out possible low-level confounds. Concurrent eye tracking confirmed that subjects accurately tracked the moving target; we were unable to decode the illusion trajectory using eye position measurements recorded during fMRI scanning, ruling out explanations based on differences in oculomotor behavior. Our results provide evidence for a perceptual representation in human visual cortex that incorporates extraretinal information.

**\*For correspondence:**
noah.steinberg@nih.gov (NJS);
zvi.roth@gmail.com (ZNR)

[†]These authors contributed equally to this work

**Competing interest:** The authors declare that no competing interests exist.

## Editor's evaluation

This important and elegant imaging experiment in humans shows that visual area hMT+, but not other candidate brain areas, signal the perceived motion path in a visual drift illusion. Using a convincing computational decoding approach, the results indicate a perceptual representation of the illusory position in space for moving stimuli even when the actual retinal position of the stimulus is kept stable. Such a representation and the underlying neural mechanisms are of broad importance for our understanding of the neural basis of sensory perception.

## Introduction

Neurons throughout visual cortex encode the location of visual stimuli on the retina, suggesting that the visual system uses a primarily retina-centered reference frame. Yet visual perception is stable across frequent eye movements that displace the retinal image. This observation has led to the idea that the brain maintains a world-centered or 'spatiotopic' representation that is invariant to changes in eye position. This idea has perhaps received its strongest support from monkey single-unit recording studies showing that neurons in the ventral intraparietal area (VIP) in macaque monkeys exhibit receptive fields that do not change position when the eyes move (*Duhamel et al., 1997*). Observations of spatiotopic representation have also been reported in a number of human brain imaging studies. For

example, visually evoked responses in both human MT/MST (hMT+) and the lateral occipital complex (LOC) have been reported to be invariant to changes in eye position (***d'Avossa et al., 2007***; ***McKyton and Zohary, 2007***). These brain imaging studies suggest a broad agreement in the brain's representation of space in monkey and human visual cortex.

Spatiotopic encoding has not been observed in all fMRI studies, however. For example, a number of studies have measured spatial receptive fields for a range of eye positions and found that receptive fields change position when the eyes move, suggesting that the brain uses a retinotopic reference frame (***Gardner et al., 2008***; ***Golomb and Kanwisher, 2012***; ***Merriam et al., 2013***). The discrepancy between these studies and reports of spatiotopic representations have not been fully resolved. One suggestion is that the reference frame for stimulus encoding depends on cognitive or task demands. For example, ***Crespi et al., 2011*** reported that the reference frame of visual responses can shift from retinotopic to spatiotopic depending on the attentional state of the observer. Behavioral studies have reported that spatiotopic representations become more prominent in tasks requiring sequences of eye movements, suggesting that spatiotopic coordinates are built-up over time (***Poletti et al., 2013***; ***Sun and Goldberg, 2016***). Together, these observations suggest that reference frames can be dynamic and depend on a variety of factors, such as visual context or the specific task (***Steinberg et al., 2022***).

In the current study, we used a version of the double-drift illusion to investigate a fundamental paradox of spatiotopic visual processing. The double-drift illusion occurs when a combination of local motion and an orthogonal global motion trajectory causes a strong perception of illusory drift

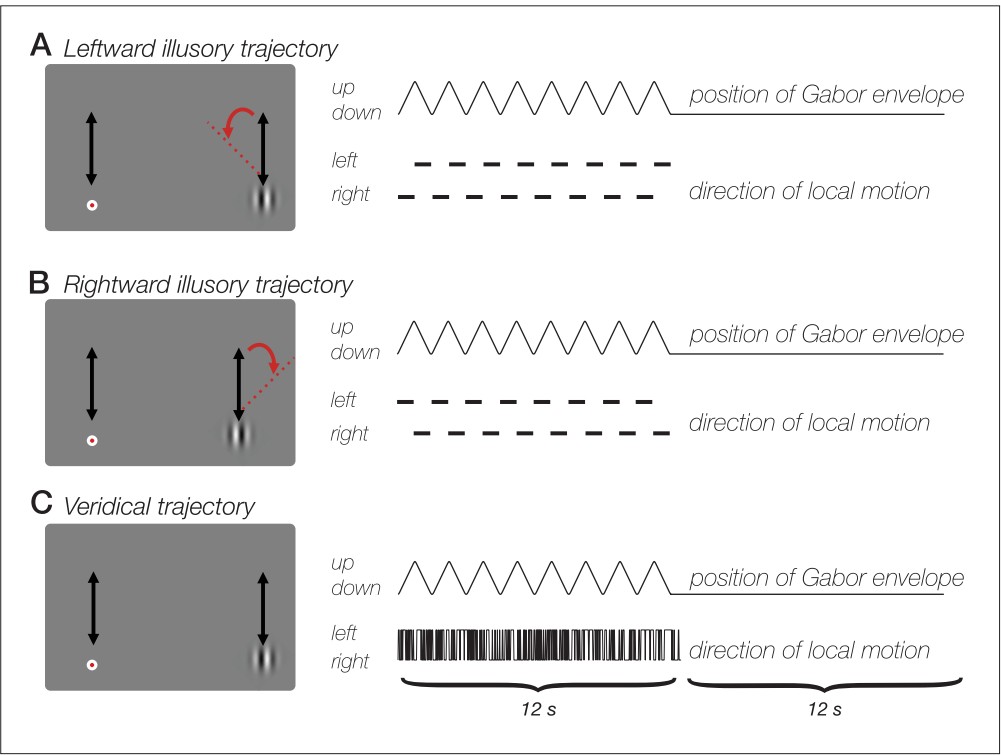

**Figure 1.** Double-drift illusion during smooth pursuit. (**A**) Leftward drift illusion. Participants made smooth pursuit eye movements, tracking the target as it moved vertically in tandem with a Gabor stimulus. Both the gabor and target moved for 12 seconds. Conjunction of local motion (grating phase drift) and global motion (displacement of the Gaussian envelope) produces an illusion in which the Gabor appears to drift several degrees to the left of its actual trajectory, even when smooth pursuit eye movements stabilize the Gabor on the retina. (**B**) Rightward drift illusion. Conjunction of local and global motion produces illusion of a rightward Gabor trajectory. (**C**) No-illusion control condition. Randomly updated grating phase does not produce illusory stimulus trajectory. All three stimulus conditions contain the same net motion energy and involved the same pursuit eye movements, yet are associated with strongly different percepts.

The online version of this article includes the following figure supplement(s) for figure 1:

**Figure supplement 1.** Mean number of voxels in each region of interest, for each experiment.

away from the veridical trajectory. The illusion can be strikingly large so that the stimulus appears to deviate by as much as 45° away from the veridical motion path (*Tse and Hsieh, 2006*; *Shapiro et al., 2010*; *Lisi and Cavanagh, 2015*). A recent study revealed that the illusion persists even during smooth pursuit when the stimulus is stabilized on the retina (*Cavanagh and Tse, 2019*). This pursuit version of the double-drift illusion highlights the paradox of spatiotopic processing: even though the stimulus is at a constant position on the retina, it is perceived to change position in world-centered coordinates. Here, we asked if this illusion could provide insight into spatiotopic encoding in the brain. We hypothesized that several regions in occipital and parietal cortex are involved in computing the illusory percept. A number of brain areas encode stable stimulus position during pursuit eye movements (i.e., 'real position' cells) (*Nau et al., 2018*). Moreover, several of these areas have been implicated in spatiotopic processing (*d'Avossa et al., 2007*). If this hypothesis is correct, we predict that activity in extrastriate cortex will reflect the illusory motion path instead of the veridical stimulus path.

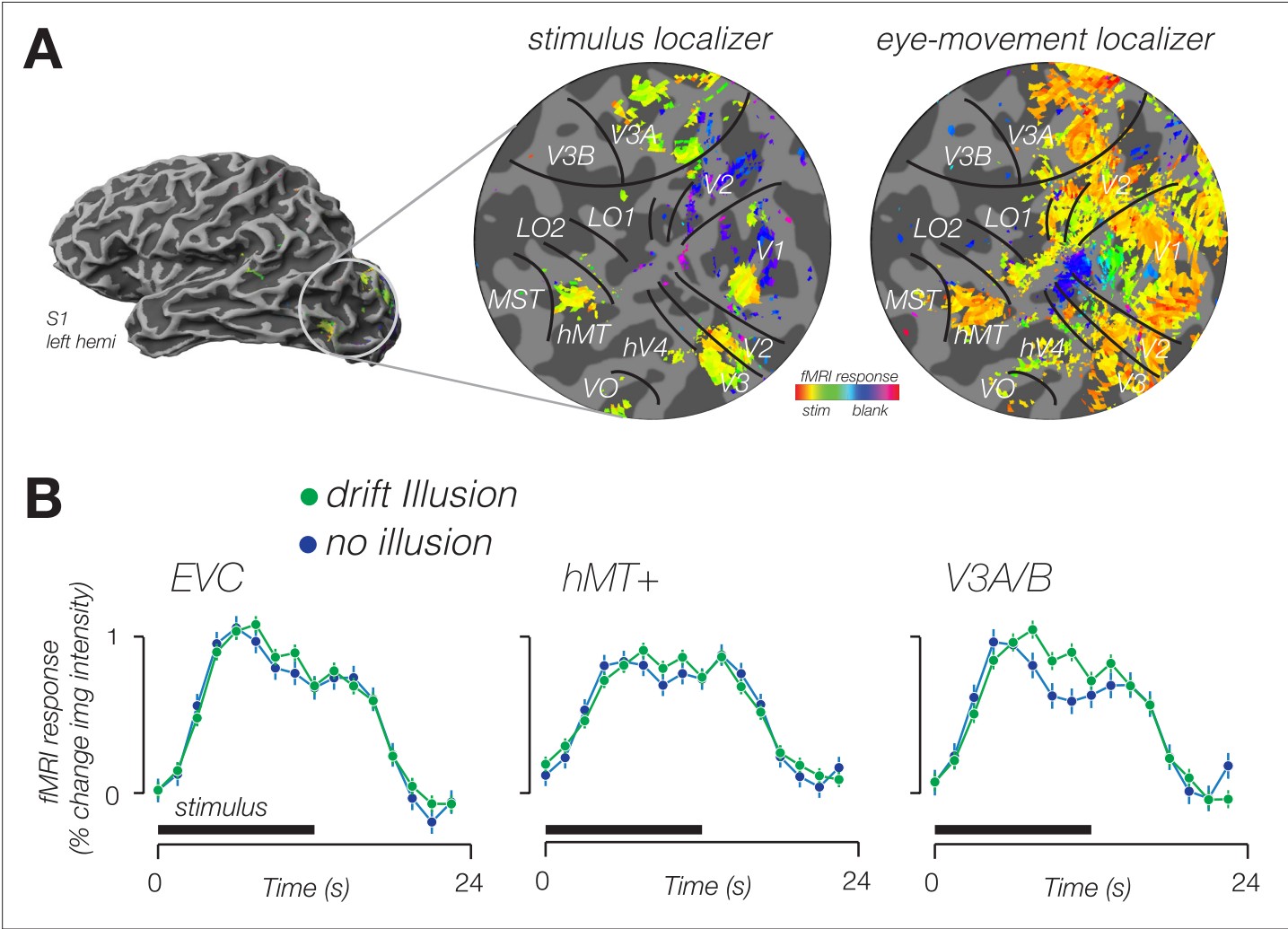

**Figure 2.** Modulation of fMRI response amplitude during double-drift illusion. (**A**) Stimulus localizer-evoked activity in cortical regions representing stimulus location (center), eye movement localizer-evoked diffuse activity in visual cortex, extending well beyond stimulus representation. Data from a single participant in the stimulus localizer shown on an inflated cortical surface (left) and a flattened patch of the occipital lobe (center). Data from the same subject in the eye-movement localizer shown on the right. Boundaries of retinotopic visual areas identified according to an anatomical template. Color indicates the phase of the response. Yellow hues indicate a response in phase with the onset of the stimulus (center) or onset of smooth pursuit (right). (**B**) Time course of fMRI response from voxels identified in the stimulus localizer, from three cortical areas exhibiting a larger response for the double-drift illusion than during a no-illusion control condition.

# Results

We tested whether fMRI BOLD activity in human visual cortex reflects the perceived spatial position of a visual stimulus that remained at a constant retinal location. We measured BOLD activity during a version of the double-drift illusion in which the perceived location of the stimulus differed from its actual location by several degrees (*Figure 1*).

To determine whether BOLD activity contained information about the visual illusion, we trained a classifier to decode blocks of illusory trials from blocks of trials in which no illusion was perceived. Using leave-one-run-out cross-validation, we found that responses in multiple cortical areas were sensitive to the double-drift illusion (*Figure 2*); the classifier could accurately decode illusory drift in all four visual area regions of interest (ROIs) (*Figure 3A*, left).

A number of different factors could lead to accurate decoding of the double-drift illusion. One possibility is that the decoder was sensitive to neural activity related to computing the location of the stimulus. Alternatively, it is possible that the perception of the illusion attracted spatial attention, and the classifier was picking up on attentional differences between illusory and non-illusory conditions. To control for this second possibility, we repeated the experiment, but had participants perform a demanding task at fixation that required sustained attention (*Haladjian et al., 2018*). The fixation task minimized differences in spatial attention to the stimulus across conditions. We again tested whether the classifier could discriminate the double-drift illusion from the control condition. While overall decoding accuracy was slightly reduced in this experiment, we found that decoding accuracy remained robust and significant in LO, hMT+, and V3A/B, but not in early visual cortex (EVC) (*Figure 3A*, right), consistent with other recent observations (*Liu et al., 2019*; *Ho and Schwarzkopf, 2022*). Participants were not attending the stimulus; therefore, these results cannot be attributed to differences in spatial attention. Instead, we conclude that the classifier was sensitive to information related to encoding the perceived position of the stimulus during the illusion.

The critical test in this study is whether BOLD fMRI activity in visual cortex can discriminate between different illusory paths. We tested whether a classifier could decode the drift path of the illusion. Of all the visual areas tested, only area hMT+ could discriminate leftward from rightward illusory paths (*Figure 3B*, left). Because the stimulus remained at a constant retinal location, the ability to discriminate the illusory motion path suggests a non-retinotopic representation of stimulus position.

We next tested alternative explanations for the ability to discriminate motion trajectory in MT+. It is conceivable that decoding of the drift path was due to subtle differences in smooth pursuit eye movements, rather than encoding of the stimulus position. Specifically, we wondered if perceiving the illusion caused a change in oculomotor behavior, which could in turn result in decodable differences in fMRI activity. Under this alternative explanation, the ability to decode the trajectory of the illusion would be a secondary consequence of any difference in oculomotor behavior between illusory conditions. We conducted the following analyses to rule out this explanation. First, we repeated the classification analysis, this time using only voxels that were selective for smooth pursuit eye movements, as identified in a separate pursuit control experiment. In this analysis, we specifically excluded voxels that responded in the stimulus localizer (see 'Smooth pursuit control experiment'). We reasoned that voxels that responded in the pursuit localizer should be most sensitive to any differences in pursuit eye movements in the main experiment. Note that this logic should apply, regardless of whether these voxels are selective for pursuit eye movements, or to the visual consequences of retinal slip during pursuit (i.e., during catch-up saccades). We found that responses in these voxels do not carry information that distinguishes the drift paths, in any of the ROIs (*Figure 3B*, right). Results from this control analysis suggest that the information being utilized by the classifier is not due to differences in pursuit eye movements. Second, in a subset of subjects, we repeated the fMRI experiment but with concurrent eye tracking (see 'Eye tracking data with concurrent fMRI'). In this subset of subjects, we replicated our main fMRI results (decoding the illusion trajectory from hMT+ responses), but we were unable to decode the illusory trajectory from the eye position measurements alone, indicating that there was no information contained in the oculomotor behavior related to perceiving the illusion (*Figure 4*). Moreover, we quantified microsaccade characteristics (amplitude and direction) and found no reliable differences between illusory conditions (illusion vs. no-illusion), or between the direction of the illusion (left vs. right). We conclude that oculomotor behavior was unlikely an underlying cause of the fMRI findings reported here.

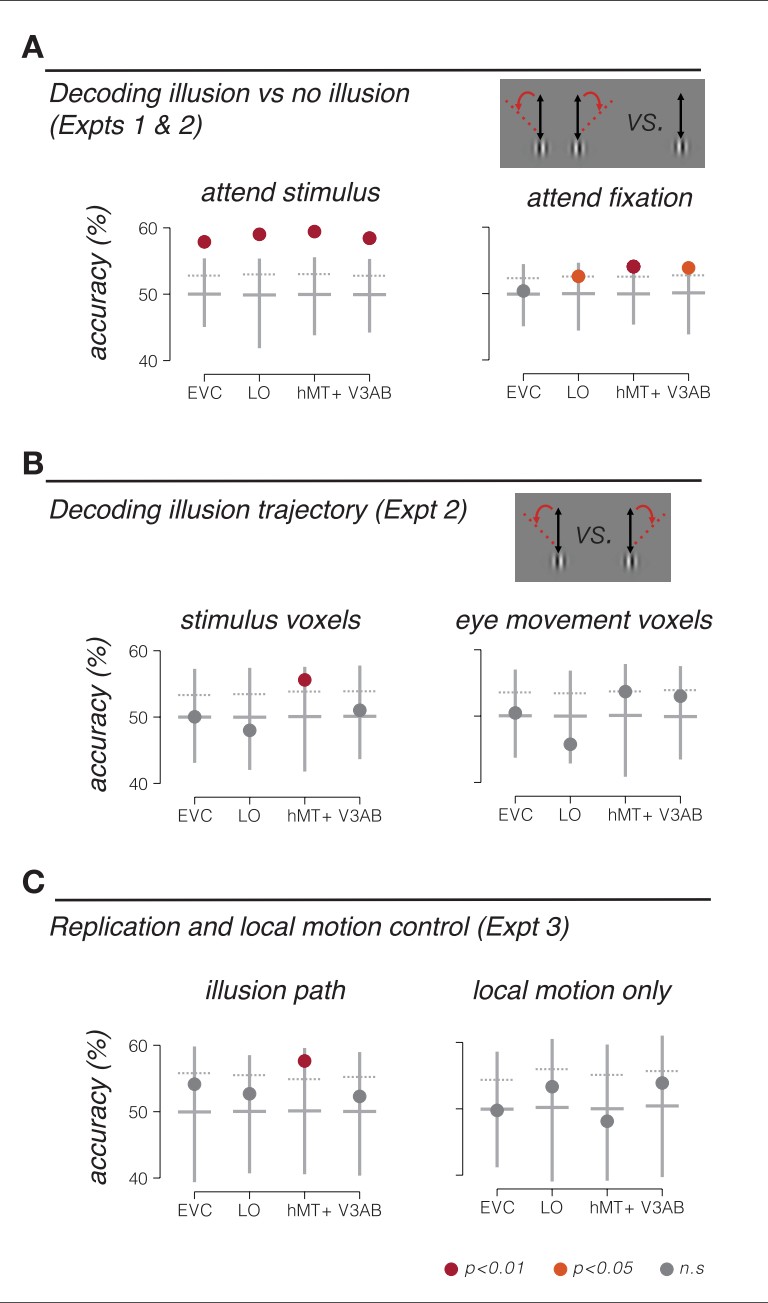

**Figure 3.** Stimulus location information encoding during double-drift illusion. (**A**) Accuracy of discriminating the double-drift illusion from a control condition that was matched for net motion energy. Participants either attended the peripheral stimulus and reported the presence of the illusion (Expt 1, left), or attended the fovea and reported a luminance decrement at fixation (Expt 2, right). (**B**) Accuracy of discriminating rightward vs. leftward drift illusion paths in Expt 2 (attend fixation) based on fMRI responses in voxels selected to match the retinotopic location of the stimulus (left) and voxels selected based on responses to pursuit eye movements (right). (**C**) Decoding accuracy for independent replication and control experiments (Expt 3). Left, decoding illusory drift paths, replicating results of Expt 2. Right, decoding local-motion only control conditions, which did not produce a drift illusion. Vertical lines extend from minimum to maximum bootstrap decoding accuracy. Horizontal lines denote median bootstrap decoding accuracy. Maroon dot, p<0.01; orange dot, p<0.05; gray dot, nonsignificant (p>0.05).

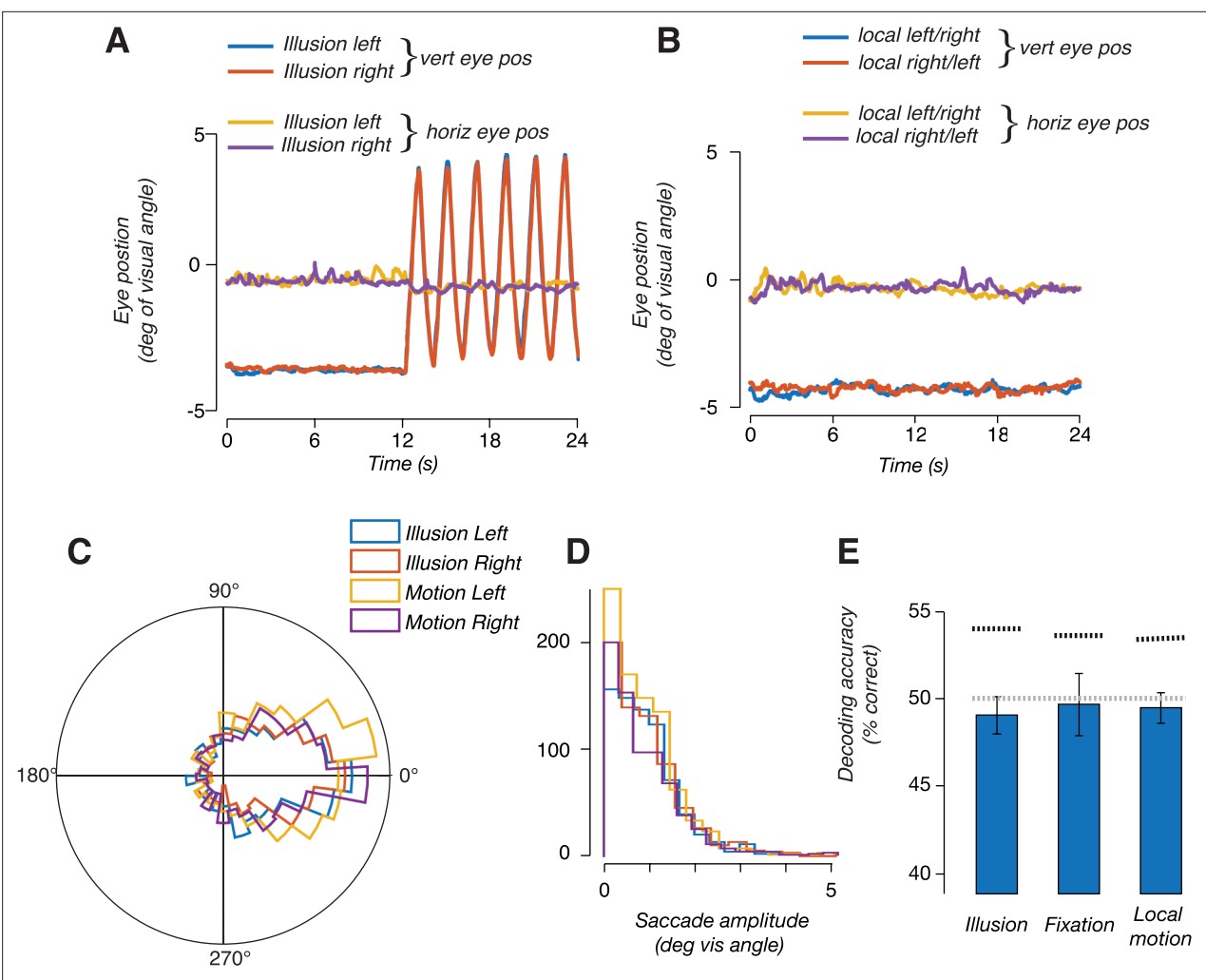

**Figure 4.** Eye position measurements did not reflect the trajectory of the illusion. (**A**) Eye position measured during blocks of double-drift illusion; one representative subject averaged over all blocks in a scan session. Traces show stable fixation during the first 12 s followed by 12 s of vertical smooth pursuit. Eye position did not differ between leftward and rightward illusion. (**B**) Eye position during blocks of local motion only trials. (**C**) Polar histogram of microsaccades direction during leftward and rightward double-drift illusion and the two motion conditions. (**D**) Histogram of saccade amplitude during the two illusion conditions and the two motion conditions. (**E**) Decoding accuracies, using the horizontal and vertical eye position measurements to train and test a linear classifier to discriminate the direction of local motion. The bar labeled 'illusion' indicates accuracy for decoding trajectory during the illusion; the bar labeled 'fixation' indicates decoding during fixation (when no illusion was perceived); bar labeled 'local motion' indicates decoding during the local-motion only condition (during fixation). Horizontal gray dashed line denotes chance decoding for binary decision (50%). Horizontal black dashed line denotes 95% confidence interval of null distribution estimated using a permutation test.

While the two illusory drift paths in our experiment were carefully balanced for net motion energy (i.e., a combination of a vertical global trajectory and horizontal local motion), we wondered if the ability of the classifier to discriminate leftward and rightward illusion drift paths could be due to the difference in temporal sequence of events within the trial (e.g., leftward followed by rightward motion, and vice versa). To test this possibility, we scanned another group of participants in an experiment (Expt 3) in which we included both illusion conditions from Expt 2, and two control conditions that contained the same local motion, but no global trajectory (and no smooth pursuit eye movements). For the illusory double-drift conditions, we again found that drift path was decodable in hMT+, replicating the results from Exp 2 in an independent group of participants (*Figure 3C*, left). However, the classifier was unable to decode the conditions containing local motion alone (i.e., discriminating left-followed-by-right from right-followed-by-leftward; *Figure 3C*, right). This result demonstrates that information about illusory motion paths in hMT+ is not due to local motion of the stimulus alone.

# Discussion

We found that fMRI BOLD responses in several visual cortical areas could reliably discriminate the double-drift illusion from a control condition that was matched for motion energy. In EVC, this result could be explained by attentional effects associated with perceiving the illusion, since when attention was directed away from the illusion, decoding in EVC dropped to chance. Beyond early visual cortex, several areas (hMT+, LO, and V3A/B) exhibited significant decoding of the illusion itself, even when controlling for spatial attention. Moreover, responses in hMT+ could also discriminate the illusory drift path, suggesting that retinal and extraretinal information are integrated in hMT+ and used to construct the spatiotopic perception experienced during the illusion. A number of control experiments indicate that these findings cannot be attributed to low-level stimulus or oculomotor factors. Our results may indicate non-retinal stimulus position encoding occurs in human extrastriate visual cortex.

## Source of illusory drift path information

What is the source of decodable drift path information? One possibility is related to a coarse-scale map for direction of motion, which has been observed throughout visual cortex, including all of the areas included in our study (*Wang et al., 2014*). The coarse-scale map for direction of motion in early visual areas (V1/V2/V3) is thought to result from an aperture-inward bias: larger responses were observed for motion away from the aperture edge (*Wang et al., 2014*). In contrast, the coarse-scale map observed in hMT+ did not depend on the aperture boundary, but instead consisted of a bias for motion toward the fovea (*Wang et al., 2014*). Could this fovea-centered bias explain the ability to decode the path of the double-drift illusion? In the current study, a fovea-centered bias would predict a leftward preference across voxels within hMT+ since the stimulus was always in the right visual field and leftward motion would be toward the fovea. However, the two illusory conditions (*Figure 1A and B*) had identical net amounts of leftward and rightward local motion, and identical proportions of time of leftward and rightward illusory drift paths. We think it is hence unlikely that a net motion bias toward the fovea in hMT + accounts for the observed results.

An alternative account is that differences in BOLD activity to the two illusory drift paths arise because of the topographic organization within hMT+ (*Huk et al., 2002*; *Amano et al., 2009*). The rightward drift path begins with illusory drift up-and-to-the-right, which increases the perceived eccentricity of the Gabor (*Lisi and Cavanagh, 2015*). The path continues with drift down-and-to-the-left, which brings the perceived position back to the original position. This cycle repeats throughout the block of trials. In contrast, the leftward drift path begins with illusory drift up-and-to-the-left, which decreases the perceived eccentricity of the Gabor, and continues with drift down-and-to-the-right, bringing the perceived position back to the original position. Thus, the average eccentricity of the perceived drift path is higher during rightward drift and lower during leftward drift. This shift in the perceived eccentricity of the stimulus could result in slightly different patterns of activity in hMT+, and this difference could underlie the ability to decode the illusion drift path.

This second account depends on there being an explicit representation of the perceived position of a stimulus in hMT+, while position encoding in EVC is entirely veridical. Since veridical position did not differ for rightward and leftward drift paths; the classifier was unable to decode the drift path from activity in EVC. Consistent with this account, one fMRI study (*Maus et al., 2013*) has reported that BOLD activity in hMT+ reflects the illusory position during motion-induced position shift. Activity throughout visual cortex is known to encode stimulus position in retinal, not spatiotopic, coordinates (*Gardner et al., 2008*). However, it remains unknown whether retinotopic coding is also universal in visual cortex for motion illusions. If the second account is accurate, our data may imply a difference between EVC and downstream areas in the spatial encoding of illusory motion.

## Spatiotopic coordinates in visual cortex

The double-drift illusion results from combining local motion of the Gabor with a global trajectory of the envelope. In the smooth-pursuit variant of the illusion, the envelope only has a trajectory when defined in spatiotopic coordinates, since the stimulus remains at a constant retinal location. With a stable position of the stimulus on the retina, the presence of the illusion suggests some degree of spatiotopic processing in the brain (*Turi and Burr, 2012*). This could be accomplished by the formation of an explicit spatiotopic reference frame (*Duhamel et al., 1997*; d'*d'Avossa et al., 2007*; *Crespi et al., 2011*). Alternatively this could be accomplished through a computation by which a retinotopic

input is combined with an eye position gain field (*Merriam et al., 2013*). Our data do not speak to which of these two possibilities is more likely.

### Decoding the drift illusion beyond hMT+

In addition to decoding the illusory drift path, patterns of activity in multiple visual areas enabled classification of the perception of the illusion. When subjects were attending the stimulus, illusion decoding was possible in all the visual areas that we studied, raising the possibility that the illusion attracted attention, resulting in a higher BOLD response during illusory blocks (see *Figure 2*). When subjects were attending to a task at fixation (Exp 2), the illusion could still be decoded from activity in LO, V3A/B, and hMT+, but not from EVC. Previous fMRI studies have claimed that the locus of attention can affect the apparent reference frame in which a stimulus is encoded (*Crespi et al., 2011*). It is unclear, however, whether attention and task indeed change the spatial reference frame, or instead affect global response amplitudes (*Roth et al., 2020*), which may constitute an additive signal obfuscating measurement of the underlying reference frame. In Exp 1, subjects performed a task on the stimulus, and so attention was likely directed toward the stimulus. In an earlier fMRI study on the double-drift illusion (*Liu et al., 2019*), subjects also performed a task on the stimulus, and while the tasks were different in the two experiments, in both cases attention was focused on the stimulus. Our results in Exp 2, in which subjects attended fixation, and not the stimulus, demonstrate that attention and task do have an impact on the spatial encoding of the double-drift illusion, and highlights the importance of controlling the attentional state of the observer when studying visual reference frames (*Crespi et al., 2011*).

### Disentangling spatiotopic representations and remapping

Two potential mechanisms have been suggested for the visual system's ability to preserve a stable percept across saccades. The first is a spatiotopic representation, relying on afferent signals that update across saccades. This can be thought of as a combination of two representations: a retinotopic representation of the visual world, and a representation of gaze direction in the world. The two are integrated to form our perception of the outside world, independent of changes in direction of gaze. The second mechanism is remapping. During (or a brief moment prior to) a saccade, receptive fields shift to where they will naturally be positioned after the saccade. This shift, or remapping, ensures that neurons activity before and after the saccade will reflect the same region in the visual field. After the saccade, the receptive field returns to its natural retinotopic position. Behavioral investigations into mechanisms for visual stability across eye movements have found evidence that both spatiotopic representations and receptive field remapping underly visual stability (*Poletti et al., 2013*), with the relative contribution of each mechanism depending on the number of intervening saccades. After a single saccade, receptive field remapping is the primary mechanism underlying visual stability, whereas spatiotopic representations become prominent after multiple saccades (*Sun and Goldberg, 2016*). It is therefore possible that fMRI studies exploring spatiotopic representations could in fact probe retinotopic coding that is updated by remapping across saccades.

The version of the double-drift illusion employed in the current study did not require saccadic eye movements, making it unlikely that perisaccadic remapping contributed to our results. Remapping can take place during saccades since saccades are discrete events separated in time, leaving time for both the shift and the return. However, shifting the receptive field cannot be used for continuous gradual changes such as smooth pursuit. A receptive field shift in the direction of the planned motion before the beginning of the pursuit would not correct for the rest of the pursuit, and furthermore, there would be no opportunity to shift back. We find it plausible to assume, therefore, that the remapping mechanism is relevant only for saccades. Note that subjects may perform saccades during the pursuit (e.g., catch-up saccades) that could be corrected by remapping, but the pursuit itself cannot be corrected by remapping. Therefore, a spatiotopic signal robust to smooth pursuit provides evidence for a different correction mechanism, namely a spatiotopic representation. From this, our results suggest that stimulus position was encoded in a spatiotopic representation.

### Relationship to a previous study of the double-drift illusion

A recent study *Liu et al., 2019* used a decoding approach to identify brain activity reflecting the percept during a version of the double-drift illusion that did not include smooth pursuit. A classifier

was used to decode both the veridical direction of a diagonally moving Gabor patch, and the illusion direction during the double-drift illusion. Briefly, they found that both veridical motion and illusory motion direction could be decoded from visual cortex, but a classifier trained on veridical motion could not decode illusory motion and vice versa, suggesting differences between the patterns of activity in the two conditions. Instead, cross-decoding was possible primarily in prefrontal cortex, suggesting that activity in PFC reflects the perceived motion direction.

The results reported by *Liu et al., 2019* are surprising, on several accounts. First, when the double-drift experiment was repeated with exactly the same stimuli, the brain regions that showed significant decoding changed (compare their Figure 4A with their Figure 6A). Second, when the veridical motion stimuli were changed slightly, the pattern of regions supporting decoding changed substantially (compare their Figure 4B with their Figure 6B), as did the regions supporting cross-decoding between veridical motion and illusory motion (compare their Figure 4C with their Figure 6C). These findings raise important questions and suggest that multiple factors, such as spatial attention, may influence the ability to decode an illusory motion path, as we have demonstrated in our study.

Regardless, the results of Liu et al. do not have direct bearing on which reference frame was used to encode the stimulus location, which is the topic of the current study. Because in Liu et al. subjects were fixating on the Gabor, the encoding of the illusion could have been in either retinal or spatiotopic coordinates. In contrast, in our study, the stimulus must have been encoded in spatiotopic coordinates. However, one potentially interesting extension of the cross-decoding approach would be to train the decoder on a version of the illusion involving fixation (as in Liu et al.), but then test the decoder on the illusion during pursuit (as in the current study). If perceived motion direction is represented in spatiotopic coordinates in both cases, one would expect the classifier to succeed in cross-decoding. However, if spatiotopic coding is used during pursuit (as we have shown here) but not during fixation, this cross-decoding should fail.

## Materials and methods

### Participants

Data were acquired from 19 healthy participants (11 females, age range 23–34 y, mean 25.8 y) with normal or corrected-to-normal vision. Experiments were conducted with the written consent of each observer. The consent and experimental protocol were in compliance with the safety guidelines for MRI research and were approved by the Institutional Review Board of the National Institutes of Health. Of the 19 participants, 12 were scanned in multiple sessions and in multiple experimental conditions. 9, 12, and 5 participants participated in Exp 1, 2, and 3, respectively.

### Stimuli

A Gabor pattern, consisting of a vertically oriented sinusoidal grating (spatial frequency of 1 cycle/°) within a Gaussian envelope (standard deviation of 1 dva), moved back and forth along a linear, vertical trajectory with a length of 8°. The trajectory length was varied slightly (±1°) across subjects to accommodate the restricted field of view within the scanner. The Gabor moved according to a linear velocity profile (10°/s) that was smoothed slightly at the top and bottom of the Gabor's path where it changed direction to facilitate accurate pursuit eye movements. The Gabor's internal grating moved orthogonally relative to its trajectory with a speed of 6.66 Hz, reversing its direction at the two endpoints of the trajectory. Participants fixated a target (0.2 dva, white dot with a black outer rim) that was positioned 9–12 dva to the left of the Gabor envelope and moved smoothly alongside it. Participants were instructed to pursue the target. Pursuit accuracy was confirmed during the behavioral experiment for each subject prior to the fMRI experiment.

### Pre-scan behavioral experiment

Prior to the first scanning session, participants viewed the double drift stimulus in a behavioral experiment and were asked to judge the angle of the illusion. Participants viewed the illusion in 12 s blocks while pursuing a target that moved smoothly alongside the stimulus. The Gabor completed eight traversals per block. At the end of each block, the Gabor was replaced with a vertical line, aligned to the physical (vertical) path of the Gabor's trajectory. Participants manipulated the angle of the line with the keyboard in order to match the perceived path of the Gabor.

## Eye tracking data with concurrent fMRI (Expt 4)

Eye tracking concurrent with 7T fMRI scanning was performed in a replication of Expt 3. We reran the experimental same experimental protocol described for (Expt 3, see below) with simultaneous eye tracking (Eyelink, 1000 Hz) on five new participants. The raw eye traces were corrected for missing data during blinks and mean centered, but not further preprocessing was performed (i.e., the eye traces were not temporally smoothed). We then used a support vector machine (SVM) classifier to the azimuth and elevation data (concatenated together to produce two vectors) together using the cosmoMVPA toolbox (*Oosterhof et al., 2016*). Decoding was attempted over trajectories/traversals of the stimulus (6 traversals × 8 runs = 48 exemplars). Three separate classifiers were designed: first, we trained and tested a classifier on eye data during the fixation portion of the trials with the visual illusion where we did not expect the traces to discriminate between conditions. This analysis served as a control. Second, we trained and tested a classifier using eye data from the illusion trials, just as we did using the fMRI data. We reasoned that any differences in the horizontal or vertical displacement of eye position during the illusion could be used to support accurate decoding. Therefore, to isolate the effect of local motion alone, we built a third classifier using eye position during the trials with only local motion. For all tests, a permutation test (1000 instances) was performed to determine the 95th percentile. Any differences in the horizontal or vertical displacement of eye position during the illusion could be used to support accurate decoding. Therefore, to isolate the effect of local motion alone, we built a classifier using eye position during the trials with only local motion. For all tests, a permutation test (1000 instances) was performed to determine the 95th percentile.

All subjects invariably make small saccades during smooth pursuit eye movements. These are typically catch-up saccades that correct for small inaccuracies in pursuit gain. It is possible that the illusion changed saccadic characteristics, such as the number, size, or direction of saccades. We therefore performed an analysis on the magnitude and direction of saccades. Saccades were typically small and did not differ in any between any of the four experimental conditions, two illusory conditions (left and right) and the two local motion-only conditions. A two-way ANOVA (illusion-vs-no-illusion × direction) was performed across all subjects and did not reveal any significant differences or interactions.

## Experimental conditions and fMRI design

### Expt 1: Attend to drift illusion path

The main fMRI experiment consisted of a randomized blocked design with three stimulus conditions, all of which involved the same vertical Gabor envelope trajectory: (1) perceived leftward drift path (internal local motion leftward during upward trajectory; rightward during downward trajectory); (2) perceived rightward drift path (internal local motion rightward during upward trajectory; leftward during downward trajectory); and (3) no-illusion control condition (randomized internal local motion, updated at 60 Hz). In all three conditions, participants pursued a fixation dot that moved smoothly and predictably alongside the Gabor, so that the Gabor remained at a constant retinal location throughout the experiment. Under conditions 1 and 2, the Gabor's perceived drift path differed from its actual trajectory by several degrees. In condition 3, the Gabor's perceived path matched its actual trajectory (i.e., there was no illusion). Each of the three conditions contained the same global and local motion energy and required the same smooth pursuit eye movements.

Each Gabor traversal lasted for 1.5 s (750 ms up, 750 ms down). The Gabor completed eight traversals in a 12 s block. The three conditions were randomly interleaved, and experimental blocks alternated with 12 s blocks of fixation in which both the pursuit target and the Gabor remained stationary, with no internal motion. Participants were instructed to press one of three buttons at the end of each experimental block indicating the direction of the illusion ('1' for leftward drift illusion, '2' for no illusion, and '3' for rightward drift). The double-drift illusion is typically strong and unambiguous, even during smooth pursuit eye movements. Accordingly, participants performed the attend-to-stimulus task with nearly 100% accuracy. Each fMRI run lasted for 288 s and included four blocks of each of the three conditions in a randomized order.

### Expt 2: Attend to fixation target

Experimental design and stimuli were identical to Expt 1, except for additional luminance decrements of the fixation target. Participants' task was to press a button when they detected the brief (250 ms) luminance decrement. Luminance decrements were determined using an adaptive, 1-up-2-down

staircase procedure (*Levitt, 1971*) prior to scanning, producing a detection rate of approximately 70% outside the scanner. We found that subjects behavior improved inside the scanner and detected the luminance changes with greater than 90% accuracy. The Gabor stimulus was not relevant to the task and subjects were not instructed to attend to it. Each fMRI run lasted for 288 s and included four blocks of each of the three conditions in a randomized order.

### Expt 3: Local motion control

This experiment controlled for differences in the pattern of local internal motion in Expts 1 and 2. While the two illusory drift paths (leftward drift, rightward drift) were balanced for net local and global motion energy, they differed in the order of motion direction. In the leftward drift illusion condition, local motion started leftward (for 750 ms) and was followed by rightward motion (for 750 ms). Vice versa for the rightward drift illusion condition. Expt 3 consisted of 4 conditions. Conditions 1 and 2 were identical to the two illusory conditions in Expts 1 and 2. However, in this experiment there were two additional control conditions in which participants viewed the same patterns of local motion as in conditions 1 and 2, but the Gabor did not move across the screen (no global motion trajectory), nor were there smooth pursuit eye movements. Instead, participants fixated a stationary target alongside a stationary Gabor containing internal motion to the left and right. In condition 3, the order of local motion was the same as in condition 1 (leftward followed by rightward). Condition 4 matched the local motion of condition 2. Participants were instructed to press a button when they detected the brief (250 ms) luminance decrement. Each fMRI run lasted for 288 s and included 3 blocks of each of the 4 conditions in randomized order.

### Expt 4: Concurrent eye tracking

This experiment was identical to Expt 3, except for two important differences. First, the experiment was performed with concurrent high-resolution eye tracking. Second, we ran the Luminance decrements task using an adaptive, 1-up-2-down staircase procedure (*Levitt, 1971*) for an extended period of time inside the scanner before the start of the experiment. Once the luminance decriment thresholds were stable at 70% inside the scanner, we fixed the decrement step size for the rest of the experiment. Analysis of behavioral performance inside the scanner confirmed that performance was as intended for each of the six subjects was as intended (66, 70, 78, 80, 75, and 71%). This was important for ensuring high task demands, so that subjects could not simultaneously attend the Gabor stimulus.

### Stimulus-only localizer experiment

In each scanning session, participants were scanned in a stimulus-only localizer experiment in which the stimulus appeared and disappeared in a two-condition block alternation protocol (9 s on, 9 s off; 14 blocks per fMRI run, lasting 252 s). Participants maintained fixation on a stationary target. A vertical Gabor stimulus appeared at the same size and eccentricity as in the main experiment. The Gabor contained internal local motion that changed direction randomly every 250 ms. After 9 s, the stimulus disappeared and participants continued to fixate. Three stimulus-only runs were included in each scanning session, one run at the beginning of the session, one in the middle, and a third run at the end of the session. Participants did not perform a behavioral task during the stimulus-only experiment.

### Eye-movement localizer experiment

In each scanning session, participants were also scanned in an eye-movement localizer experiment in which participants tracked a moving fixation dot that was identical to the double-drift illusion experiments, except that there was no peripheral Gabor stimulus. Responses to pursuit eye movements were measured in a two-condition block alternation protocol (9 s pursuit, 9 s fixation; 14 blocks per fMRI run, each lasting 252 s). Two eye-movement localizer runs were included in each scanning session, one run at the beginning of the session and a second run at the end of the session. Participants did not perform a behavioral task during eye-movement localizer experiment.

## Experimental setup: Behavioral

Stimuli were generated using MATLAB (MathWorks, MA) and MGL (*Gardner et al., 2018*) on a Macintosh computer, and presented on a 61-inch screen (BenQ XL242OZ) positioned 57 cm away from the

participant. Participants were seated in a darkened room and were head stabilized by a chin rest. An Eyelink 1000 eye-tracking system was used to measure binocular eye position at 1000 Hz. Eye-tracking calibration was performed at the beginning of the session and repeated intermittently throughout the session to ensure that eye tracking accuracy remained within 1° of visual angle throughout the experiment.

## Experimental setup: fMRI

Stimuli were generated using MATLAB (MathWorks) and MGL (*Gardner et al., 2018*) on a Macintosh computer. Stimuli were displayed via a PLUS U2-1200 LCD projector (resolution: 1024 × 768 pixels; refresh rate: 60 Hz) onto a back-projection screen in the bore of the magnet. Participants viewed the display through an angled mirror at a viewing distance of approximately 58 cm, producing a field of view of 20.5° × 16.1°.

fMRI data were acquired from participants on a research-dedicated Siemens 7T Magnetom scanner using a 32-channel head coil, located in the Clinical Research Center on the National Institutes of Health campus (Bethesda, MD). Functional imaging was conducted with 56 slices oriented parallel to the calcarine sulcus covering the posterior half of the brain: TR: 1500 ms; TE 23 ms; FA: 55°; voxel size: 1.2 × 1.2 × 1.2 mm with 10% gap between slices; grid size: 160 × 160 voxels. Multiband factor 2, GRAPPA/iPAT factor 3. The slices covered all of the occipital and parietal lobes, and the posterior portion of the temporal lobe. For Expt 5, voxel size was increased to 1.8 × 1.8 × 1.8 mm, with 0% gap, Multiband factor 2, and GRAPPA/iPAT factor 2.

For each participant, a high-resolution anatomy of the entire brain was acquired by co-registering and averaging between 2 and 8 T1-weighted anatomical volumes (magnetization-prepared rapid-acquisition gradient echo, or MP2RAGE; TR: 2500 ms; TE: 3.93 ms; FA: 8°; voxel size: 0.7 × 0.7 × 0.7 mm; grid size: 256 × 256 voxels). The averaged anatomical volume was used for co-registration across scanning sessions and for gray-matter segmentation and cortical flattening. Functional scans were acquired using T2*-weighted, gradient recalled echo-planar imaging to measure blood oxygen level-dependent (BOLD) changes in image intensity (*Ogawa et al., 1990*). The in-plane anatomical was aligned to the high-resolution anatomical volume using a robust image registration algorithm (*Nestares and Heeger, 2000*).

Prior to the first experimental functional run of each session, 30 volumes were acquired with identical scanning parameters and slice prescription as the subsequent functional runs, except for the phase encoding direction which was reversed. This single reverse phase-encoded run was used to estimate the susceptibility-induced off-resonance field using a method similar to that described in *Andersson et al., 2003* as implemented in FSL (*Smith et al., 2004*). This estimate was then used to correct the spatial distortions in each subsequent run in the session.

## fMRI preprocessing and analysis

The anatomical volume acquired in each scanning session was aligned to the high-resolution anatomical volume of the same participant's brain using a robust image registration algorithm (*Nestares and Heeger, 2000*). Head movement within and across scans was compensated using standard procedures (*Nestares and Heeger, 2000*). The time series from each voxel was divided by its mean to convert from arbitrary intensity units to percent modulation and high-pass filtered (cutoff = 0.01 Hz) to remove low-frequency noise and drift (*Smith et al., 1999*).

## ROI definition

ROIs were defined according to an anatomical template (*Benson and Winawer, 2018*). Such templates are inherently imprecise because of inter-subject variability. Each subject has a slightly different anatomy and a slightly different location of functional ROIs. The smaller an ROI the more severe this imprecision becomes relative to the ROI size. To mitigate this problem, we combined nearby regions to form larger ROIs which we analyzed. V1, V2, and V3 were combined to form an early visual cortex ROI (EVC); LO1 and LO2 were combined into LO; V3A and V3B were combined into V3A/B; TO1 and TO2 were combined to form hMT+, corresponding to MT and MST.

## Data analysis for stimulus-only and eye-movement localizers

The three stimulus localizer scans were averaged together, and the two eye-movement scans were averaged together. The first cycle of each averaged time series was discarded, leaving 13 cycles. Each

individual voxel's time course was fitted to a cosine with a period matching the cycle duration of 12 volumes (18 s). Each voxel was then assigned the correlation coefficient and phase of the best-fitting cosine. Voxels with a correlation coefficient greater than 0.2 were considered active in the localizer.

## GLM analysis

BOLD fMRI time series were averaged across all voxels within each ROI. Trials were then divided into two conditions: double-drift illusion (either leftward or rightward) and no illusion. Each condition was modeled by 16 predictors, one for each time point in the 24 s following the beginning of the trial. Deconvolution was performed by multiplying the pseudoinverse of the condition predictor matrix with the time series (*Dale, 1999*). This procedure yields two hemodynamic response functions for each ROI. The average of the two response functions was used as the ROI's hemodynamic response function for the next analysis step. Next, a design matrix was constructed for each ROI, with a single HRF function modeling each 12 s block. Response amplitudes were then computed by taking the pseudoinverse of this design matrix and multiplying it with the single voxel time series.

The goal of this analysis was to estimate a response amplitude for each voxel on each scanning run. The first step was to estimate a single hemodynamic response function for each ROI and each participant. This was accomplished by averaging across all voxels within the ROI, and then using deconvolution (*Gardner et al., 2008*) to estimate a hemodynamic response function (collapsing across the different conditions) over a 24 s period following the beginning of the block. Next, this single hemodynamic response was used to create a design matrix, treating each of the conditions independently. Response amplitudes were then computed by taking the pseudoinverse of this design matrix and multiplying it with the time series for each individual voxel within the ROI. This procedure allowed for differences in the shape of the hemodynamic response across different ROIs and across different participants.

## Decoding analysis

In multivariate classification analysis of fMRI data, each condition is represented by a set of points in multidimensional space, with dimensionality equal to the number of voxels and each point corresponding to a single measurement. Accurate decoding is possible when the responses corresponding to different conditions form distinct clusters within this high-dimensional space (*Pereira et al., 2009*). We measured the amplitude of the fMRI response during 12 s blocks of trials, in which each block consisted of eight up-down traversals of the double-drift illusion. We took the beta weight from the GLM analysis (see above) as the amplitude of the response during each 12 s block as a single input to the classification analysis. These response amplitudes were stacked across blocks within a run, and across runs within a session, forming an m × n matrix, with m being the number of voxels in the region of interest and n being the number of repeated measurements in the session. The value of n was typically 64 (for 14 of 19 participants), and ranged from 48 (1 participant) to 96 (4 participants). We only included voxels with a GLM $R^2$ in the top 50th percentile. Prior to decoding each voxel's beta weights were z-scored. Decoding was performed with a maximum likelihood classifier using the MATLAB function 'classify' with the option 'diagLinear' (*Roth et al., 2018*). Decoding accuracy was computed using leave-one-run out cross-validation. The m × n data matrix was partitioned along the n dimension (repeated measurements) into training and testing sets, in which the training set consisted of the blocks from all but one of the runs, and the testing set included the blocks (from all three conditions) from the left out run. Because the data in the training and testing sets were drawn from different runs in the same session, they were statistically independent. The training set was used to estimate the parameters (multivariate means and variances) of the maximum-likelihood classifier. The testing set was then used for decoding. Decoding accuracy was determined as the proportion of the test examples that the classifier was able to correctly assign to one of the two illusory drift paths. Illusion decoding was performed separately for left vs no-illusion, and for right vs no-illusion, and then averaged across both.

The leave-one-run-out cross-validation procedure resulted in a single decoding accuracy estimate per ROI per session. A non-parametric permutation test was used to evaluate the significance of this decoding accuracy. Specifically, we constructed a distribution of accuracies expected under the null hypothesis that there is no difference between the two illusory drift paths. To generate a null distribution decoding accuracy, we permuted the block labels for each run and repeated the

leave-one-run-out decoding analysis. Repeating this randomization 1000 times yielded a distribution of accuracies expected under the null hypothesis. Accuracies computed using the unrandomized training data were then considered statistically significant when decoding accuracy was higher than the 95th percentile of the null distribution (p<0.05, one-tailed permutation test).

## Acknowledgements

This work was supported by the Intramural Research Program of the National Institutes of Health (ZIAMH002966), National Institute of Mental Health Clinical Study Protocol 93M-0170, NCT00001360.

We would like to thank Francisco Pereira, June Kim, and Cassie Jones for assisting with this study.

---

## Additional information

### Funding

| Funder | Grant reference number | Author |
| --- | --- | --- |
| National Institute of Mental Health | ZIAMH002966 | Elisha Merriam |

The funders had no role in study design, data collection and interpretation, or the decision to submit the work for publication.

### Author contributions

Noah J Steinberg, Conceptualization, Formal analysis, Investigation, Methodology, Writing – original draft, Writing – review and editing; Zvi N Roth, Data curation, Formal analysis, Supervision, Investigation, Visualization, Methodology, Writing – original draft, Writing – review and editing; J Anthony Movshon, Conceptualization; Elisha Merriam, Conceptualization, Resources, Data curation, Software, Formal analysis, Supervision, Funding acquisition, Investigation, Methodology, Project administration, Writing – review and editing

### Author ORCIDs

Noah J Steinberg ⓘ http://orcid.org/0000-0003-1337-1479
Zvi N Roth ⓘ https://orcid.org/0000-0002-2173-1625
J Anthony Movshon ⓘ https://orcid.org/0000-0002-0274-9213
Elisha Merriam ⓘ https://orcid.org/0000-0003-2787-566X

### Ethics

Clinical trial registration ClinicalTrials.gov identifier: NCT00001360.
All participants granted informed consent under an NIH Institutional Review Board approved protocol (93-M-0170, ClinicalTrials.gov identifier: NCT00001360).

### Decision letter and Author response

Decision letter https://doi.org/10.7554/eLife.76803.sa1
Author response https://doi.org/10.7554/eLife.76803.sa2

---

## Additional files

### Supplementary files

• Transparent reporting form

### Data availability

Data and code are publicly available on Github (https://github.com/elimerriam/doubleDrift copy archived at *Merriam, 2024*) and the Open Science Framework (https://doi.org/10.17605/OSF.IO/3ZX8P).

The following dataset was generated:

| Author(s) | Year | Dataset title | Dataset URL | Database and Identifier |
|---|---|---|---|---|
| Roth ZN, Steinberg NJ, Merriam EP | 2024 | Data and code for "Brain representations of motion and position in the double drift illusion" | https://doi.org/10.17605/OSF.IO/3ZX8P | Open Science Framework, 10.17605/OSF.IO/3ZX8P |

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
