## [Editor Report]

This important and elegant imaging experiment in humans shows that visual area hMT+, but not other candidate brain areas, signal the perceived motion path in a visual drift illusion. Using a convincing computational decoding approach, the results indicate a perceptual representation of the illusory position in space for moving stimuli even when the actual retinal position of the stimulus is kept stable. Such a representation and the underlying neural mechanisms are of broad importance for our understanding of the neural basis of sensory perception.

---

## [Decision Letter]

**Decision letter after peer review:**

Thank you very much for submitting your article "Neural Basis of The Double Drift Illusion" for consideration by *eLife*. Your article has been reviewed by 3 peer reviewers, including Kristine Krug as the Reviewing Editor and Reviewer #1, and the evaluation has been overseen by Tirin Moore as the Senior Editor. The following individuals involved in the review of your submission have agreed to reveal their identity: Corentin Gaillard (Reviewer #1); Maria Concetta Morrone (Reviewer #2).

Essential revisions (for the authors):

There are three main areas, the reviewers have identified that need further clarification and some limited additional analyses. They are summarised here. Please use the recommendations to the authors below for the more detailed questions that need addressing.

1) Motivation of the particular study and discussion of the results in relation to the existing literature needs expansion for the reader to be able to better evaluate the specific contribution of this particular study.

- The authors motivate the study by saying that there have been conflicting results about which brain areas are involved in spatiotopic coding, but they did not give an indication about why there might be conflicting results or why the current study is suitable to address the previous discrepancies. Is this study simply adding another observation to the existing body of literature, or does it go beyond previous studies in a critical theoretical way, especially also with regard to Liu et al. 2019 (Current Biology)?

- A more nuanced discussion of the more controversial literature on retinotopic vs. spatiotopic visual coding and where the current work is situated in introduction and discussion. The authors seem to confirm the importance of attention, and that could be made more explicit. In addition, there is much evidence both for retinotopy and spatiotopy which is simply overlooked.

- How do the current results relate to the literature on a role of hMT+ in global motion perception? Is this a potential alternative interpretation of the results and if not, why not.

2) The results and methods require more detailed explanation and some limited additional data, in particular with regards to:

- the eye movement controls

- the ROI definitions

- the decoding method

- some of the more marginal statistical results

3) The reviewers would like to see a direct, quantitative comparison of the decoding for different motion directions of the drift illusion with and without attention (i.e. the attention comparison in 3A but for decoding left-right trajectory).

*Reviewer #1 (Recommendations for the authors):*

1) lines 289-298: Eye movement control: "Pursuit accuracy was confirmed during the behavioral experiment or each subject prior to the fMRI experiment." A weakness of the paper is the lack of eye movement measurements during the MR scan. Changes in eye movement for the combination of different combinations of up- and downwards drift with directions of motion could potentially provide an explanation of the source of the perceptual signals.

(i) The authors should report the detailed behavioural data from their eye movement controls. For instance, show up and down pursuit trajectories separately when combined with left or right stimulus motion to exclude eye movement drift as a signal source and test whether they are different.

(ii) This control is also relevant to the last point from the discussion (line 256) whether the authors are dealing with (a) a retinotopic representation associated with saccade driven remapping of receptive field or (b) a spatiotopic representation.

The authors defend the second argument based on the fact that their task does not require saccades (therefore no remapping), but there could still be catch-up saccades during the smooth pursuit phase and there is also no control for micro-saccades.

2) The authors want to rule out the hypothesis that differences in pursuit eye movements could account for the decoding performance. In figure 3B right, they claim that hMT eye movement voxels did not contain information about the illusionary trajectory but:

Accuracy for that decoding is 0.53+-0.8, p-value 0.052, which is just not significant. If you look at the table, For Expt 1, second condition, LO based decoding is at 0.5260 +- 0.10 and has p-value of 0.047. This could arise from the fact that the decoded categories are not the same and from changes in 95%CI, but given the number of comparisons and very close results, Expt 2 is a little bit less convincing than either Exit 1or 3 with their elegant designs.

This requires a more detailed and differentiated discussion of the underlying statistics.

The authors should also show (in a supplement) the distribution of the eye movement related voxels they excluded/analysed separately in the different subjects.

3) In a previous (neurophysiology) study, the authors made the point that V5/MT signals in macaques were not coding a global motion percept (Hedges et al. 2011). A critical point there was the size of stimulus relative to the receptive field size in V5/MT. In reference to that, a more detailed discussion would be helpful, in terms of the extent to which the receptive fields in hMT+ encode spatial position in this paradigm and the size of their stimulus relative to hMT+ receptive field sizes might shed light on the underlying neural mechanism.

In general, it would be good to see more discussion of how the current study is situated relative to other studies that suggested (or not) a role of V5/MT with regard to perceptual signalling.

4) More experimental detail in some areas would be helpful to the reader (see also points 1-3)

- Abstract: Include type details of analysis used.

- eye movement data from behavioural experiments (supplement)

- number and distribution of eye movement related voxels.

*Reviewer #2 (Recommendations for the authors):*

One major concern, which could confuse readers, is the treatment of previous research of the authors. The opening line is: "The primate visual system is retinotopic: neurons throughout visual cortex encode the location of visual stimuli on the retina (Gardner et al., 2008)": a simple statement of undisputed fact, with no qualification, no mention that several other papers, before and after Gardner, have reported different data and drawn different conclusions. Later in the introduction we hear that there is indeed a controversy, with a possible explanation for the discrepant results (attentional focus). Later still their results are presented, strongly supporting a non-retinotopic representation in hMT+. Then in the discussion, the first sentence is reiterated: "Activity throughout visual cortex is known to encode stimulus position in retinal, not spatiotopic, coordinates (Gardner et al., 2008)", with no discussion of the discrepancy between the current results and this simple, undisputed "historical fact".

This is very hard to follow. However much one is attached to one's own work, it cannot be considered Gospel truth and everything else noise. This becomes particularly bizarre when the "noise" is consistent with the current research, and the previous publication not. I think the readers deserve a discussion on the discrepancy, and how best to move forward. The authors seem to confirm the importance of attention, and that could be made more explicit (but see methodological criticisms). In addition, there is much evidence both for retinotopy and spatiotopy which is simply overlooked.

Perhaps the title could be improved, to reflect the actual conclusions of the paper (non-retinal motion response in MT). It does not really discover the "Neural basis for the double-drift illusion", as it specifically examines only the condition when observers track the stimulus, stabilizing it on the retina. It also does not speak to the main result, spatiotopic cortical representation.

The results and methods presentation also requires a more detailed explanation. At present many analyses are unclear or use the wrong methodology.

1. Definition of ROIs. It is unclear if the stimuli used to define the map of figure 2A is moving vertically and if the %BOLD of figure 2B is taken for the extended ROI defined by the atlas as stated in the methods, or are simply the average of the significant foci of figure 2A. Given the maximum activity of 1% I am inclined to say that it is the response of only the significant voxels in the ROIs. Usually, peripheral stimuli do not produce such large activity. But if so the analysis is subject to circularity!

In the methods they state that given the great variability of segmentation with atlas they pooled together early visual areas. This is very peculiar, given the much greater variability in the segmentation of LO1 and LO2. Why not use a better and complete atlas, like the Glasser? Why not show a map of the average across all subjects?

2. The decoding methods are very compressed and many details are not available. Normally for any decoding strategy the average activity is normalize so the decoding is not biased by the different mean. Has this been done?

Decoding responses to stimuli that have a different average power of signal is trivial, so the results of figure 3A that compare two stimuli with greatly different energy should be eliminated. In any case the different stimulus power between stimuli in Figure 1C and the other two should be measured and discussed.

3. The attention control is important, given the previous dispute, but it should be run between the two different motion directions. The question is: in the unattended condition can the direction of the illusory motion been decoded?

4. Also eye-movements is a crucial aspect, but not so much for pursuit given that hopefully the subject were fixating, but for drift. The direction of eye drift could bias the decoding results. However, the comparison of decoding between sets of voxels of different numerosity is a potential problem. The stimulus voxels should be reduced in size to match the other ROI.

To conclude, I think the work is interesting and important, worth eventual publication in a good journal. However, it needs a major rewrite, detailing better important technical details, reviewing existing literature with a less egocentric bias, and discussing better the apparent conflicts between this paper and previously published studies, including those of the authors.

*Reviewer #3 (Recommendations for the authors):*

Recommendations

1. Regarding major comment 3 -- I believe there are a couple of things the authors can do to increase confidence that eye movements are not driving the results. (1) Redo the decoding from stimulus selective voxels while excluding all pursuit selective voxels. (2) If it is not possible to measure (or ideally, control) eye position in the scanner, a behavioral version of the "attend to fixation" experiment could be performed with eye tracking to determine whether smooth pursuit eye movements are affected by the illusory drift direction.

2. It would be helpful if the authors could more thoroughly report the behavioral results for both tasks, including performance levels and threshold values on the fixation task to demonstrate that the task was demanding and the staircase was working well.

---

## [Author Response]

Essential revisions (for the authors):There are three main areas, the reviewers have identified that need further clarification and some limited additional analyses. They are summarised here. Please use the recommendations to the authors below for the more detailed questions that need addressing.1) Motivation of the particular study and discussion of the results in relation to the existing literature needs expansion for the reader to be able to better evaluate the specific contribution of this particular study.- The authors motivate the study by saying that there have been conflicting results about which brain areas are involved in spatiotopic coding, but they did not give an indication about why there might be conflicting results or why the current study is suitable to address the previous discrepancies. Is this study simply adding another observation to the existing body of literature, or does it go beyond previous studies in a critical theoretical way, especially also with regard to Liu et al. 2019 (Current Biology)?

Here, the Reviewers are making two separate points. The first point regards the earlier controversy regarding whether or not human area MT contains an explicit map-like representation of absolution spatial position that is invariant to eye position (i.e., a spatiotopic representation, rather than a retinotopic representation). We have revised the text in several places to summarize this literature and better contextualize our results given the previous work in this field.

The second point regards the novelty of our results in light of a previous study on the double drift illusion (Liu et al., 2019). Liu et al. addressed a different question. In that study, subjects fixated for the entire duration of the experiment, so the issue of visual reference frame was not addressed; Liu et al. instead focused on the representation of stimulus position during fixation (regardless of reference frame) and found significant decoding in far anterior frontal regions. We have now clarified this important distinction in the manuscript.

- A more nuanced discussion of the more controversial literature on retinotopic vs. spatiotopic visual coding and where the current work is situated in introduction and discussion. The authors seem to confirm the importance of attention, and that could be made more explicit. In addition, there is much evidence both for retinotopy and spatiotopy which is simply overlooked.

We have addressed this comment by substantially editing the Discussion.

- How do the current results relate to the literature on a role of hMT+ in global motion perception? Is this a potential alternative interpretation of the results and if not, why not.

Please see our response below to Reviewer 1, Comment 1ii.

2) The results and methods require more detailed explanation and some limited additional data, in particular with regards to:- the eye movement controls- the ROI definitions- the decoding method- some of the more marginal statistical results

We address each of these points in considerable detail below where they appear in the Reviewers comments.

3) The reviewers would like to see a direct, quantitative comparison of the decoding for different motion directions of the drift illusion with and without attention (i.e. the attention comparison in 3A but for decoding left-right trajectory).

Please see response to Reviewer 2 below.

Reviewer #1 (Recommendations for the authors):1) lines 289-298: Eye movement control: "Pursuit accuracy was confirmed during the behavioral experiment or each subject prior to the fMRI experiment." A weakness of the paper is the lack of eye movement measurements during the MR scan. Changes in eye movement for the combination of different combinations of up- and downwards drift with directions of motion could potentially provide an explanation of the source of the perceptual signals.

We now include eye tracking data.

We have addressed the Reviewer's comment by repeating the fMRI experiment in a new group of subjects in which we were able to also obtain concurrent, high-quality eye tracking. When we initially conducted the experiment, it was not possible to perform eye tracking in the 7T scanner at NIH. Because of this limitation, we were forced to depend on careful eye tracking in a pre-scan behavioral experiment. But in the ensuing period of time, we have developed a protocol for obtaining high quality eye tracking with an Eyelink 1000 mounted in the bore of the scanner. Now that we have the ability to collect concurrent eye tracking, we repeated the fMRI experiment and found that our main fMRI result replicated (i.e, it was possible to decode the direction of the illusion from fMRI responses in hMT+). Additional, the concurrent fMRI eye tracking enabled us to make four important observations (see new Figure 4):

First, subjects maintained stable fixation when the target was stationary during fixation and accurately pursued the vertically moving target during illusion (Figure 4). This analysis confirms that the drifting Gabor remained at a relatively fixed position on the retina during the illusory period.

Second, there were no differences in microsaccades between any of the conditions. We quantified the direction, amplitude, and frequency of all saccades for each condition. While we did observe small rightward microsaccades, none of the microsaccade characteristics differed between conditions. The rightward microsaccades may have been due to the sustained eccentric leftward fixation. Or, it may have been due to attention to the right visual field stimulus (despite the foveal attention task). Or it may have reflected the known horizontal microsaccade bias. Regardless, we do not believe our fMRI results are related to microsaccades because these small saccades did not differ across condition.

Finally, we wondered if small not-easily-quantified ocular deviations could have differed between conditions, and somehow result in differences in fMRI activity picked up by the decoding analysis. To test for this possibility, we trained a classier to discriminate condition based on the raw eye traces (just as we did in the main fMRI data analysis). But unlike the fMRI analysis, we found that it was not possible to decode the direction of the illusion from the eye traces themselves.

We conclude that the ability to decode the illusion from fMRI responses were not due to differences in eye movements caused by the illusion.

(i) The authors should report the detailed behavioural data from their eye movement controls. For instance, show up and down pursuit trajectories separately when combined with left or right stimulus motion to exclude eye movement drift as a signal source and test whether they are different.

See comment above.

(ii) This control is also relevant to the last point from the discussion (line 256) whether the authors are dealing with (a) a retinotopic representation associated with saccade driven remapping of receptive field or (b) a spatiotopic representation.The authors defend the second argument based on the fact that their task does not require saccades (therefore no remapping), but there could still be catch-up saccades during the smooth pursuit phase and there is also no control for micro-saccades.

The reviewer correctly points out that our argument against remapping depends on there being no saccades during the illusion, yet subjects were making microsaccades and catch-up saccades during the pursuit phase of the illusion (our new eye tracking data confirm this). Could remapping during these small saccades produce the illusion? While an intriguing possibility, we think this is unlikely. The small saccades would have to be very regular and consistent to produce such a continuous percept. Nonetheless, we now discuss this possibility.

2) The authors want to rule out the hypothesis that differences in pursuit eye movements could account for the decoding performance. In figure 3B right, they claim that hMT eye movement voxels did not contain information about the illusionary trajectory but:Accuracy for that decoding is 0.53+-0.8, p-value 0.052, which is just not significant. If you look at the table, For Expt 1, second condition, LO based decoding is at 0.5260 +- 0.10 and has p-value of 0.047. This could arise from the fact that the decoded categories are not the same and from changes in 95%CI, but given the number of comparisons and very close results, Expt 2 is a little bit less convincing than either Exit 1or 3 with their elegant designs.This requires a more detailed and differentiated discussion of the underlying statistics.

Here the Reviewer makes the valid point that one cannot conclude anything from the lack of significance in a particular ROI. We completely agree. It has been convincingly demonstrated that with sufficient SNR, significant effects can be observed throughout the brain, even for the simplest tasks (Gonzalez-Castillo et al., PNAS, 2012), and hence we have little doubt that with infinite scanning, we would eventually observe significant decoding even in the pursuit-selective voxels (and elsewhere too).

We aren't claiming that regions that exhibit selectivity for smooth pursuit *don't* contain information about the illusion. Rather, we emphasize that regions that are selective for the stimulus *do* contain information about the illusion. This is important because it is precisely these stimulus-related voxels that represent the position of the stimulus during fixation, and hence should also be affected by a shift in perceived position of the stimulus during the illusion.

The authors should also show (in a supplement) the distribution of the eye movement related voxels they excluded/analysed separately in the different subjects.

We have added a flat patch of occipital cortex to Figure 2 showing the spatial distribution of both stimulus-selective and pursuit-selective voxels based on the two localizer scans for one example subject. Stimulus-selective voxels were located where one would expect based on retinotopy. Pursuit-selective voxels were generally located in more peripheral portions of the retinotopic map. Smooth pursuit voxels were also generally more spatially diffuse, with a larger number of responsive voxels. These pursuit voxels may reflect the retinal stimulation caused by the edge of the screen moving on the retina as subjects moved their gaze vertically.

3) In a previous (neurophysiology) study, the authors made the point that V5/MT signals in macaques were not coding a global motion percept (Hedges et al. 2011). A critical point there was the size of stimulus relative to the receptive field size in V5/MT. In reference to that, a more detailed discussion would be helpful, in terms of the extent to which the receptive fields in hMT+ encode spatial position in this paradigm and the size of their stimulus relative to hMT+ receptive field sizes might shed light on the underlying neural mechanism.In general, it would be good to see more discussion of how the current study is situated relative to other studies that suggested (or not) a role of V5/MT with regard to perceptual signalling.

The reviewer points out that MT has been shown to code local rather than global motion for stimuli in which there were both local and global motion that were distinct from each other. Such distinction between local and global motion existed in our stimuli where the local motion was horizontal and the global motion was vertical. We interpret the reviewer’s comment to suggest that perhaps global motion signaling was involved in processing the double drift stimulus. However, we point out that the global motion component was the same for all illusory conditions. Hence, we think it unlikely that mechanisms that support global motion perception underlie this particular illusion.

4) More experimental detail in some areas would be helpful to the reader (see also points 1-3)- Abstract: Include type details of analysis used.

We have expanded the abstract to include many more details about the analyses.

- eye movement data from behavioural experiments (supplement)

In the revision, we now include eye movement data collected at the same time as fMRI data.

- number and distribution of eye movement related voxels.

This is now included as part of Figure 2 and in Figure 1—figure supplement 1.

Reviewer #2 (Recommendations for the authors):One major concern, which could confuse readers, is the treatment of previous research of the authors. The opening line is: "The primate visual system is retinotopic: neurons throughout visual cortex encode the location of visual stimuli on the retina (Gardner et al., 2008)": a simple statement of undisputed fact, with no qualification, no mention that several other papers, before and after Gardner, have reported different data and drawn different conclusions. Later in the introduction we hear that there is indeed a controversy, with a possible explanation for the discrepant results (attentional focus). Later still their results are presented, strongly supporting a non-retinotopic representation in hMT+. Then in the discussion, the first sentence is reiterated: "Activity throughout visual cortex is known to encode stimulus position in retinal, not spatiotopic, coordinates (Gardner et al., 2008)", with no discussion of the discrepancy between the current results and this simple, undisputed "historical fact".This is very hard to follow. However much one is attached to one's own work, it cannot be considered Gospel truth and everything else noise. This becomes particularly bizarre when the "noise" is consistent with the current research, and the previous publication not. I think the readers deserve a discussion on the discrepancy, and how best to move forward. The authors seem to confirm the importance of attention, and that could be made more explicit (but see methodological criticisms). In addition, there is much evidence both for retinotopy and spatiotopy which is simply overlooked.

We recognize that we were too selective in our review of the literature on retinotopic organization in the early visual pathway, and we have consequently reframed both the Intro and Discussion to acknowledge that extra retinal factors can modulate the retinotopic maps that are routinely observed in early visual cortex.

Perhaps the title could be improved, to reflect the actual conclusions of the paper (non-retinal motion response in MT). It does not really discover the "Neural basis for the double-drift illusion", as it specifically examines only the condition when observers track the stimulus, stabilizing it on the retina. It also does not speak to the main result, spatiotopic cortical representation.

We have changed the title to be more closely aligned with the result: ***Brain representations of motion and position in the double drift illusion*.**

The results and methods presentation also requires a more detailed explanation. At present many analyses are unclear or use the wrong methodology.1. Definition of ROIs. It is unclear if the stimuli used to define the map of figure 2A is moving vertically and if the %BOLD of figure 2B is taken for the extended ROI defined by the atlas as stated in the methods, or are simply the average of the significant foci of figure 2A. Given the maximum activity of 1% I am inclined to say that it is the response of only the significant voxels in the ROIs. Usually, peripheral stimuli do not produce such large activity. But if so the analysis is subject to circularity!

We have clarified in the Methods how voxels were selected. Briefly, we used a conjunction of an anatomical atlas and functional localizers. The two functional localizers (the ‘stimulus-alone’ and ‘pursuit-alone’ conditions) were completely independent of the main experiment (different runs, different experimental protocol), so none of the analyses were circular in any way. This has all now been clarified in the revision.

In the methods they state that given the great variability of segmentation with atlas they pooled together early visual areas. This is very peculiar, given the much greater variability in the segmentation of LO1 and LO2. Why not use a better and complete atlas, like the Glasser? Why not show a map of the average across all subjects?

The Reviewer is referring to our explanation in the Methods section in which we explained why we combined multiple Atlas-defined ROI's into a larger, more general ROI. Fundamentally, the issue has to do with the spatial accuracy of all of the human cortical atlases that currently exist. All atlases, including the Glasser atlas, involve the registration of individual subject data to a template. There are spatial errors that are inherent in this procedure — and again, that’s true of all atlases. The spatial errors largely stem from the variability across subjects in sulcal anatomy and the degree to which functional areal boundaries follow anatomic landmarks. These factors result in a fundamental limit in the accuracy of any atlas to accurately identify a particular cortical area. To circumvent this limitation, we simply pooled voxels across adjacent atlas-defined ROIs in V1, V2, and V3 to make a general ‘early visual cortex’ ROI, and TO1 and TO2 to make a single hMT+ ROI. We have published with both ROI criteria previously (e.g., Roth et al. *PLOS Bio*, 2021; Burlingham et al., *eLife*, 2022).

2. The decoding methods are very compressed and many details are not available. Normally for any decoding strategy the average activity is normalize so the decoding is not biased by the different mean. Has this been done?

We have added a number of important details to the methods section that were indeed lacking in the initial submission. For example, we now describe that stimulus-related and eye-movement voxels were identified by localizer coherence > 0.2. (*Methods: Data analysis for stimulus-only and eye-movement localizers)*. For decoding, we only used voxels with R^2^ in top 50^th^ percentile. Betas were z-scored for each voxel. (*Methods: Decoding analysis*).

Some fMRI studies using MVPA subtract off the average activity across the ROI prior to building and testing the decoder. But we feel this approach is not appropriate because including the mean cannot bias decoding toward one condition or another (as long as the experimental design is well balanced, which was the case in our experiment). Moreover, subtracting the mean can, in fact, lower sensitivity to real effects. For example, decodable information may be present in the ROI mean response amplitude, or it may be present in the spatial pattern of activity across voxels. Both results are valid, but analyses pipelines involving mean-subtraction are only sensitive to the latter.

Decoding responses to stimuli that have a different average power of signal is trivial, so the results of figure 3A that compare two stimuli with greatly different energy should be eliminated. In any case the different stimulus power between stimuli in Figure 1C and the other two should be measured and discussed.

We don't agree with the reviewer's assertion that the stimuli have different average power. They are matched for contrast energy, which is the way that we and most others think about the 'power' of the stimulus. We are, of course, studying a motion representation — we therefore had to manipulate *motion* energy; it is the core of our experiment. It may not have been clear to the reviewer why we designed the experiment this way. But we contend that there was no other way to design the stimulus while still producing the illusion.

3. The attention control is important, given the previous dispute, but it should be run between the two different motion directions. The question is: in the unattended condition can the direction of the illusory motion been decoded?

We ran the analysis that the reviewer suggested (attempting to decode the direction of the illusion when subjects were not performing a detection task on the pursuit target). In this condition, we found no reliable information about the direction of the illusion. This is somewhat surprising. One explanation for this is that the attentional control in Expt 2-4 served to stabilize attentional state, thereby improving the overall signal-to-noise ratio of the measurement. In Expt 1, subjects simply reported whether or not they perceived the illusion, and hence we cannot be certain of where their attention was directed. We conclude that uncontrolled attention may have obfuscated the signal needed to train the decoder to discriminate the illusion trajectory.

4. Also eye-movements is a crucial aspect, but not so much for pursuit given that hopefully the subject were fixating, but for drift. The direction of eye drift could bias the decoding results.

Please see our response above. In a newly added experiment with eye tracking during the fMRI task, we now show that we cannot decode the different conditions from the eye traces themselves (Figure 4).

However, the comparison of decoding between sets of voxels of different numerosity is a potential problem. The stimulus voxels should be reduced in size to match the other ROI.

We agree with the Reviewer's points regarding ROI size. As Figure 2 now shows, there were many more voxels in the eye movement localizer. And yet, we were unable to decode the direction of the illusion from these ROIs. Hence, we do not think that the ability to decode from the stimulus-selective voxels is due to differences in ROI size.

To conclude, I think the work is interesting and important, worth eventual publication in a good journal. However, it needs a major rewrite, detailing better important technical details, reviewing existing literature with a less egocentric bias, and discussing better the apparent conflicts between this paper and previously published studies, including those of the authors.

We thank the reviewer for the helpful comments and hope that the revision addresses these points satisfactorily.

Reviewer #3 (Recommendations for the authors):Recommendations1. Regarding major comment 3 -- I believe there are a couple of things the authors can do to increase confidence that eye movements are not driving the results. (1) Redo the decoding from stimulus selective voxels while excluding all pursuit selective voxels. (2) If it is not possible to measure (or ideally, control) eye position in the scanner, a behavioral version of the "attend to fixation" experiment could be performed with eye tracking to determine whether smooth pursuit eye movements are affected by the illusory drift direction.

To be certain that errors in smooth pursuit weren’t driving the results, we repeated the experiment with concurrent eye tracking in 5 subjects during 7T fMRI scanning. We attempted to decode the direction of the illusion directly from the eye traces, reasoning that any difference in oculomotor behavior (such as, for example, differences pursuit gain or catchup saccades) would enable decoding of the two conditions. We found that there were no differences in eye tracking between the two illusory conditions (see Figure 4)

We repeated the decoding analysis (as shown in Figure 3) after excluding pursuit selective voxels in hMT+. In all cases, decoding in hMT+ was still significant, albeit less robust (see our response to Major Comment 3, above). We believe this is because there were pursuit-selective voxels in hMT+ that overlap with stimulus selective voxels, which is not surprising given the large population receptive fields of voxels in hMT+ and the small size of the retinotopic map. Excluding pursuit-selective voxels yielded a smaller population with which to conduct the analysis.

2. It would be helpful if the authors could more thoroughly report the behavioral results for both tasks, including performance levels and threshold values on the fixation task to demonstrate that the task was demanding and the staircase was working well.

In addressing the reviewer’s question, we realized that performance on the attention task during scanning was much higher than we expected (>90% accuracy). We think that this may have been because we ran the staircase after putting subjects in the scanner but prior to running the main experiment, at which point we locked the stimulus parameters. Subjects may have become acclimated to the scanning environment and their performance may have thus improved. To address this issue, we reran the experiment in a new group of 5 subjects (see results in Figure 4). We ran the staircase for a much longer period of time at the start of each session until their thresholds were stable at 70% correct. In these new scan sessions, we also had concurrent eye tracking, addressing other concerns brought by the reviewers comments (discussed above). In these new data (see supp Figure 1), behavioral accuracy was on average 70% over the course of the entire session, as expected by the 1-up 2-down staircase that was run prior to the main experiment, so we are confident that subjects were directing attention to the fixation task. The main finding was replicated: we were able to decode the direction of the illusion in hMT+. And there was no decodable information about the direction of the illusion in the eye traces.